# Fabrication of Electrospun PLA-nHAp Nanocomposite for Sustained Drug Release in Dental and Orthopedic Applications

**DOI:** 10.3390/ma16103691

**Published:** 2023-05-12

**Authors:** Nishat Anzum Kanak, Md. Shahruzzaman, Md. Sazedul Islam, Makoto Takafuji, Mohammed Mizanur Rahman, Sumaya F. Kabir

**Affiliations:** 1Department of Applied Chemistry and Chemical Engineering, University of Dhaka, Dhaka 1000, Bangladesh; 2Department of Chemistry, University of Pittsburgh, Pittsburgh, PA 15260, USA; 3Department of Chemistry and Biochemistry, Florida State University, Tallahassee, FL 32306, USA; 4Department of Applied Chemistry and Biochemistry, Kumamoto University, Kumamoto 860-8555, Japan

**Keywords:** electrospun, nanocomposite, drug release, cytotoxicity, hydroxyapatite

## Abstract

This study describes the fabrication of nanocomposites using electrospinning technique from poly lactic acid (PLA) and nano-hydroxyapatite (n-HAp). The prepared electrospun PLA-nHAP nanocomposite is intended to be used for drug delivery application. A hydrogen bond in between nHAp and PLA was confirmed by Fourier transform infrared (FT-IR) spectroscopy. Degradation study of the prepared electrospun PLA-nHAp nanocomposite was conducted for 30 days both in phosphate buffer solution (PBS) of pH 7.4 and deionized water. The degradation of the nanocomposite occurred faster in PBS in comparison to water. Cytotoxicity analysis was conducted on both Vero cells and BHK-21 cells and the survival percentage of both cells was found to be more than 95%, which indicates that the prepared nanocomposite is non-toxic and biocompatible. Gentamicin was loaded in the nanocomposite via an encapsulation process and the in vitro drug delivery process was investigated in phosphate buffer solution at different pHs. An initial burst release of the drug was observed from the nanocomposite after 1 to 2 weeks for all pH media. After that, a sustained drug release behavior was observed for the nanocomposite for 8 weeks with a release of 80%, 70% and 50% at pHs 5.5, 6.0 and 7.4, respectively. It can be suggested that the electrospun PLA-nHAp nanocomposite can be used as a potential antibacterial drug carrier for sustained drug release in dental and orthopedic sector.

## 1. Introduction

Biodegradable and biocompatible composites can be used as a drug delivery system since they can control the drug dissolution rate [1]. Biodegradable polymer-based nanocomposites can be used in different configurations, i.e., nanogel, nanosphere, and nanofibers for drug delivery application [2]. However, among these configurations, electrospun nanofibers are most favorable since electrospinning is a simple and most widely used technique to fabricate nano fibers of excellent properties in a cost-efficient way [3]. Fibers produced by the electrospinning process possess porous structure, and a greater surface area to volume ratio causes higher drug encapsulation and sustainable drug delivery [4].

Among biodegradable aliphatic polyesters, poly lactic acid (PLA) is considered to be the most promising candidate as a drug carrier due to its biocompatibility, tunable degradation property, hydrophobicity, etc. [5]. Moreover, PLA has the characteristic of good formability into fibers, and PLA nanofibers mimic the structural dimension of the extracellular matrix of native tissues and organs [6].

Generally, for bone or dental implant applications, hydroxyapatite (HAp) is widely used due to its osteoconductive and osteointegration behavior [7]. Therefore, the incorporation of HAp into an implant can be used to regulate cell reactions and regenerate tissue [8]. Nano-hydroxyapatite (nHAp) possesses excellent biological efficacy due to higher solubility and therefore better biological responses. Thus, a blending of these two materials (PLA and n Hap) is a potential candidate to be used as an implant, in bone repairing and in drug delivery application. Fu et al. [9] suggested that the polyester/hydroxyapatite (polyester/HA) composites enhance bone tissue repairing since they imitate the structure of natural bone tissue.

Nanocomposites can carry the drug molecules on its surface and is a suitable material for the sustained release of drug molecules [10]. PLA-nHAp nanocomposites are gaining much interest due to their excellent biocompatibilty, tunability and nanostructure to fit in bone and tooth defects [3]. Thus, PLA-nHAp nanocomposite will act as a potential candidate for the delivery of drugs and other therapeutic agents and to treat bone infection locally. For instance, Farkas et al. [3] prepared doxycycline-loaded electrospun PLA/HAP nanofibers as a drug delivery system and reported the prolonged release of doxy.

The bone repair processes in dental and orthopedic surgery involve the risk of postoperative complications, i.e., osteomyelitis [11]. Such an infection can lead patients to prolonged hospitalization, inflammation and functional disability of the limbs, sepsis, etc. [12]. The available treatment for osteomyelitis requires high dosages of antibiotics for a significant stretch of time [12]. In this regard, treating the infection using in situ drug-release composites can be an optimal solution to prevent orthopedic post-surgical complications. An in situ drug-release composite can promote drug action by delivering sufficient doses of the pharmaceutical agent to a specific site of the body and limit further infections and side effects [13].

In this work, a biowaste eggshell was used to produce nano-hydroxyapatite (nHAp). The electrospun PLA-nHAp nanocomposite was prepared and a gentamicin sulfate was loaded to the nanocomposite through encapsulation to investigate the drug release at different pHs. Enhanced biocompatibility due to the addition of nHAp, tunable physical properties such as dimeter of fiber, thermal stability, degradation time, and drug release capacity of this composite make it a potential candidate for dental and orthopedic application.

Numerous studies have been performed on polymer-HAP nanocomposites, studying processing condition, influence of filler on polymer fiber morphology, etc. However, few reports have been devoted to studying the drug release phenomena of electrospun polymer-nHAp nanocomposite. The aim of this study is to develop a PLA and nHAp nanocomposite to use as an in situ drug carrier for dental and bone diseases treatment.

## 2. Experimental

### 2.1. Materials

The PLA used in this study was supplied by NatureWorks™, Minneapolis, MN, USA (molecular weight (M_w_) is in between 115 × 10^4^–145 × 10^4^ g/mol). To produce a nano-hydroxyapatite (nHAp), biowaste eggshell was collected from a local bakery shop. Nitric acid (HNO_3_) (69%) and monobasic potassium dihydrogen phosphate (KH_2_PO_4_) [>99.5%] were purchased from Merck, Darmstadt, Germany. Ammonium hydroxide (NH_4_OH) was purchased from Sigma Aldrich, Burghausen, Germany. Gentamicin sulfate (GS) drug was collected from Square Pharmaceuticals Limited, Gazipur, Bangladesh. All the chemicals were used without any further purification.

### 2.2. Nano-Hydroxyapatite (nHAp) Synthesis

The nano-hydroxyapatite (nHAp) was synthesized using the following procedure [14]. The biowaste eggshells were collected and the cuticle layer inside the shell was washed by water. After that, the eggshells were crushed partially to dry up easily, and placed in an oven to remove any moisture at a temperature of 100 °C for 2 h where the heating rate was 5 °C/min. After the heat treatment, a simple mill grinder was used to grind the oven-dried eggshells into finely crushed eggshell powder. Then, the fine powder was carried through calcination process in an electric furnace at 900 °C for 2 h. During this heat treatment, the calcium carbonate (CaCO_3_) of the eggshell was converted into calcium oxide (CaO) by releasing carbon dioxide (CO_2_). Thus, the derived CaO was dissolved fully using concentrated nitric acid (HNO_3_), and then potassium dihydrogen phosphate of 0.6 M was slowly added into it. Ammonium hydroxide solution was further added until the pH of the solution reached at 10, and then stirred vigorously for an hour. The final mixture was kept at room temperature for aging overnight. A double-layered solution was formed with clear ammonia solution on top and the white precipitate of nano-hydroxyapatite (nHAp) at bottom. After washing the synthesized nHAp several times with distilled water, nHAp was finally dried for 24 h.

### 2.3. Preparation of PLA Fiber

The PLA solution was prepared using the procedure described by Casasola, R. et al. [15]. A certain amount of PLA was dissolved in a binary solvent system of DMF and DCM (40/60 *v*/*v*) by a magnetic stirrer at room temperature for 2 h to obtain solutions of different concentrations % (*w*/*v*), shown in Table 1. PLA fiber was successfully drawn from two compositions (PF-4 and PF-5) using the electrospinning method.

### 2.4. Fabrication of PLA-nHAp Nanocomposite by Electrospinning

PLA-nHAp solution for electrospinning was prepared following the procedure described by Sonseca et al. [16]. A total of 3 wt% nHAp was dispersed in DCM for an hour by ultrasound agitation. PLA solution was prepared as mentioned above and the dispersed nHAp was added to the PLA solution and stirred for 2 h to obtain uniform mixing. For electrospinning, the PLA-nHAp solution was taken into the syringe and an electrode was attached to the needle. Prior to the electrospinning, air bubbles were purged from the syringe and the solutions were electrospun at 20 kV and at a flow rate of 0.6 mL/h. The rotating metal collector was kept at a distance of 10 cm from the charged needle tip. The randomly oriented electrospun fibers were collected from the rotating metal collector covered by aluminum foil. The PLA-nHAp nanocomposite (PLA:nHAp = 25:3) was successfully drawn from the following composition which was further studied for drug loading and in vitro drug releasing application.

### 2.5. Characterization Techniques

To characterize nHAp, an energy dispersive X-ray (EDX) spectrometer Hitachi SU8000, Tokyo, Japan, and a transmission electron microscope (TEM), JEM-1400 plus (JEOL), Tokyo, Japan, were used. For TEM analysis, nHAp was dispersed in ethanol by ultrasound agitation, and then a drop of dispersed solution was drop cast onto a copper grid. Filter paper was used to remove excess solution. The grid was used for TEM observation after vacuum drying. The particle size of nHAp was calculated and plotted using Image J software. To examine the morphology of the electrospun fiber, SEM analysis was carried out using JEOL6400 SEM, Japan. The viscosities of the PLA solutions were measured using a viscometer (Elcometer 2300 rv-2, Manchester, UK) at room temperature.

Fourier transform infrared spectroscopy (FT-IR) analyses of nHAp, PLA, gentamicin (GS), PLA-nHAp nanocomposite and gentamicin loaded nanocomposite (PLA-nHAp-GS) were recorded carefully in the range of 4000–400 cm^−1^ using FTIR 8400S, Shimadzu, Kyoto, Japan.

To monitor the crystalline structure of the samples and to find out the crystallite size of nHAp nanoparticles, Ultima IV X-ray diffractometer (Rigaku Corporation, Tokyo, Japan) was used with Cu-Kα radiation. Crystal size of HAp was calculated using Schrrer equation.
(1)$$d=\frac{K\lambda }{\beta \cos\theta }$$
where, d = crystallite size, λ = X-ray Wavelength (0.154 nm), K = Dimensionless shape factor (0.9), β = line broadening (in radian) at half the maximum intensity (FWHM) which is found to be 1.588° (0.0277 radian), and θ = Bragg diffraction angle which is 31.7° (0.28 radian) for nHAp.

Thermal properties of nHAp, PLA and PLA-nHAp nanocomposite were evaluated by a thermogravimetric analyzer (TGA), TGA-50H, Shimadzu, Japan, in a nitrogen atmosphere at a heating rate of 10 °C/min from room temperature to 650 °C in an alumina cell. Sample weight varies in the range from 2.25 to 6.55 mg.

### 2.6. Degradation Study

Degradation studies of the prepared PLA-nHAp nanocomposite were performed for a period of 30 days in water and in PBS at room temperature. The prepared PLA-nHAp nanocomposite (1 mm × 1 mm × 1 mm) was immersed in both demineralized water (pH 6.67) and PBS of pH 7.4 for 30 days. On a week interval, a small amount of nanocomposite was taken and dried, and FTI-R was performed.

### 2.7. Loading of Gentamicin Sulfate

An amount of 1 wt% of Gentamicin sulfate was dispersed in PLA-nHAp solution using ultrasound agitation for an hour. The prepared PLA-nHAp-GS solution was then electrospun as described before. The prepared drug-loaded nanocomposite was then collected from metal collector. The loading of GS was confirmed from FT-IR spectroscopy, and its distribution in PLA-nHAp matrix was observed by SEM analysis.

### 2.8. In Vitro Analysis

The gentamicin sulfate release profile was determined using the method described by Thakur et al. [17]. GS absorbs UV and visible light poorly, so indirect spectrophotometric method was used for assaying GS using ninhydrin as a derivatizing agent. Quantitative analysis of gentamicin was carried out using the ninhydrin aqueous solution where the ninhydrin reacted with primary and secondary amines of gentamicin, producing a purple color.

For GS calibration, 10 gL^−1^ freshly prepared ninhydrin aqueous solution was mixed with aliquot of gentamicin solution. The mixture was then heated in a water bath at a temperature of 95 ± 0.1 °C for 15 min resulting in a purple color solution. After cooling in an ice-water bath, the UV–visible spectra were measured (at wavelength of 400 nm) for both the gentamicin-ninhydrin solution and the blanks.

The drug release profile of the gentamicin-loaded PLA-nHAp-GS was studied at three different pHs at room temperature for 8 weeks. Buffer solutions of pH values 5.5, 6 and 7.4 were prepared using K_2_HPO_4_, KH_2_PO_4_, NaOH and deionized water, respectively. A GS-loaded nanocomposite was immersed in each buffer solutions (pH values of 5.5, 6 and 7.4, respectively) at room temperature. Every week, 2 mL of each buffer solution was withdrawn and the amount of GS was quantified using the procedure as stated above. The same amount of fresh phosphate buffer was used to replace the withdrawal amount from the dissolution media.

The kinetic models used for explaining drug release are zero order, first order, Higuchi and Korsmeyer–Peppas models, etc. In this study, the in vitro drug release data were fitted into these models and calculated using the following equations:

Zero-order release model:Q_t_ = K_0_ t(2)

First-order release model:Ln Q_t_ = K_1_t + Q_0_(3)

Higuchi model:Q_t_ = K_H_ t^0.5^(4)

Korsmeyer-Peppas model:Q_t_ = K_p_ t^n^(5)

In the above four equations, Q_t_ and Q_0_ denote the amounts of drug released at time t and at initial time (zero), respectively. K_0_, K_1_, K_H_ and K_p_ are the drug release constants for individual kinetic-release equations.

### 2.9. Cytotoxicity Test

Cytotoxicity analysis was carried out at the Centre for Advanced Research for Science (CARS) lab at the University of Dhaka to assess the biocompatibility of the electrospun nanocomposite. BHK-21 cells (fibroblast cell) and Vero cells (epithelial cells) were kept individually in DMEM (Dulbecco’s Modified Eagles medium), containing 10% foetal bovine serum (FBS), 1% penicillin- streptomycin (1:1) and 0.2% gentamycin. To observe the cell culture, all cells (1.5′ 104/100 mL) were seeded on 24-well plates and incubated at 37 °C + 5% CO_2_ at first. After removing the culture media, 100 μL of samples (autoclaved) and 400 μL of fresh media were added at each on the following day. Cytotoxicity was investigated for 48 h under an inverted light microscope. Further, the BHK-21 fibroblast cell was held in agar media culture to assess biocompatibility of the nanocomposite. For samples preparation, duplicate wells were used. By the aid of a digital camera, photos were taken and analyzed with commercial software.

## 3. Results and Discussion

### 3.1. Morphology and Crystal Structure of nHAp

Appendix A is representing the TEM image of nHAp. The image confirms the formation of nano-sized hydroxyapatite. Some of the nHAp particles are of irregular hexagonal shape and some are of rod-like shape. The average diameter of nHAp has been found to be uniform in the range of 7.6 nm from Appendix A.

Appendix A shows the XRD pattern of pure nHAp. From XRD, it can be said that the hydroxyapatite nanoparticles are crystalline. The main (h k l) diffraction peaks observed for nHAp are at (002), (211), (300), (130), (222), (213) and (004), and comply with the phases listed within the JCPDS database. However, a peak identified near 2θ = 25.75° is possibly due to presence of carbonate since nHAp was synthesized in an open environment [18].

### 3.2. Morphology of PLA Fiber

Figure 1 represents the SEM images of PLA fiber (PF) produced from different concentrated PLA solutions.

From Figure 1a,b, it can be suggested that the fibers count of samples PF-1 and PF-2 is very poor. Fiber formation can be seen for PF-3 (Figure 1c). Fiber formation increased significantly for PF-4 (Figure 1d) and, finally, for PF-5 (Figure 1e). A good network for fibers formed, resembling the exact fiber structure without any agglomeration. Figure 1f is representing the histogram of PF-5, where the average diameter of PLA fiber is found to be 118 nm. Since PF-5 is identified for the best fiber formation, the nanocomposite with nHAp was prepared for PF-5 only.

### 3.3. Morphology of PLA-nHAp Nanocomposite

Figure 2a represents the SEM image PLA-nHAp nanocomposite (25:3) fiber, denoting smooth, uniform and highly oriented fibers without any beads or agglomeration. From the histogram in Figure 2b, the diameter of PLA-nHAp nanocomposite fiber was found to be 260 nm.

It has been reported that the diameter of electrospun nanofibers is usually controlled by several parameters which are solution concentration, viscosity, electrospinning parameters, etc. [19]. In this study, electrospinning parameters such as voltage, working distance, flow rate, etc., were exactly the same for synthesizing both PLA fiber (PF-5) and the nanocomposite (PLA-nHAp). Since other conditions are identical, the increased diameter (260 ± 90 nm) of nanocomposites in comparison to PF-5 (118 ± 42 nm) can be attributed to the presence of nHAp in nanocomposites. Farkas et al. also reported an increased diameter for the PLA-HAP nanofiber to 310 *±* 12 nm, whereas for pure PLA the diameter was reported to be 288 *±* 11 nm.

### 3.4. FT-IR Analysis of PLA-nHAp Nanocomposite

Figure 3 represents the FT-IR spectra of pure PLA, nHAp and the prepared nanocomposite. For PLA, the characteristic bands identified at 1210 cm^−1^, 1365 cm^−1^ and 1446 cm^−1^ are due to the stretching vibrations of the C-O-C, the symmetric vibration of -CH_3_ and the asymmetric vibration of -CH_3,_ respectively [20,21]. The peak identified at 1742 cm^−1^ and 2980 cm^−1^ can be attributed to the presence of carbonyl group and the presence of C-H bond, respectively, in the main chain of PLA [21].

For nHAp, the peaks identified at 1026 cm^−1^, 858 cm^−1^ and 600 cm^−1^ can be assigned to the asymmetric stretching of PO_4_^3−^ group of nHAp [18,22]. The peaks at 1710 cm^−1^ and 3336 cm^−1^ are due to the presence of C-O bond of carbonate ion and the stretching of OH, respectively [22].

The prepared PLA-nHAp nanocomposite possesses all the characteristic peaks of both PLA and nHAp. For nanocomposite, peaks observed at 1197, 1367, 1450, 1750 and 2980 cm^−1^ resemble the characteristics peaks for PLA, and peaks at 873 and 1096 cm^−1^ resemble the peaks for nHAp. Although the PLA-nHAp nanocomposite is showing all characteristic peaks of PLA and nHAp, all peaks shifted slightly, which could be due to the interaction of inorganic nanoparticle nHAp with organic polymer PLA [23]. Moreover, in nanocomposites, the intensity of −CH_3_ absorption peaks at 1367 cm^−1^ reduced significantly in comparison to peak at 1365 cm^−1^ for PLA, which may be attributed to the interfacial bonding of PLA with nHAp, causing change in the crystal structure and a decrease in the crystallinity of PLA [23].

In nanocomposites, the peak for PO_4_^3−^ group of nHAp at 1026 cm^−1^ was also found to be increased and shifted to 1096 cm^−1^. However, in nanocomposites, the peak of nHAp at 3336 cm^−1^ (for −OH group) disappeared and the peak intensity of PLA at 1742 cm^−1^ (for C=O) decreased and shifted to 1750 cm^−1^. This observation indicates the formation of a hydrogen bond (Appendix A) between −OH group of nHAp and C=O of PLA in the nanocomposite [24].

### 3.5. XRD Studies of PLA-nHAp Nanocomposite

Figure 4 represents the XRD pattern of PLA, nHAp and the prepared PLA-nHAp nanocomposite. X-ray diffractogram of PLA presents the principal diffraction peaks of PLA at 17.2° and 19.5° and can be attributed to the α form of the PLA crystals [16]. The wide diffraction peaks in nanocomposite at 2θ = 17°–27° can be attributed to the amorphous nature of PLA in the nanocomposite. This reduction in intensity and displacement of XRD peaks of pure PLA in nanocomposite may due to the formation of interfacial bond between nHAp and PLA [25]. Moreover, in the nanocomposite, the peak positions of nHAp at 2-theta = 25.76°, 31.7° and 40.04° shifted to higher 2-theta position at 2θ = 31.9° (002), 37.8° (211) and 44.1° (130), respectively, and the intensity also decreased. These observations indicate changes in the crystal structure of nHAp after nanocomposite formation [23].

### 3.6. Thermogravimetric Analysis of PLA-nHAp Nanocomposite

The TGA thermogram presented in Figure 5 indicates that thermal decomposition of PLA was initiated at around 160 °C and completed at around 400 °C. Both the PLA and PLA-nHAp nanocomposite had shown similar thermal decomposition pattern. However, for PLA, the degradation started at around 160 °C, whereas nanocomposite fiber started to degrade at the comparatively higher temperature of ~220 °C. Moreover, the maximum degradation of PLA occurred at ~290 °C, and for nanocomposite was identified at ~310 °C. This higher thermal stability of nanocomposite can again be attributed to the bonding between the PLA and nHAp [26].

### 3.7. Degradation Study of PLA-nHAp Nanocomposite

Degradation studies of the prepared PLA-nHAp nanocomposite were observed for a period of 30 days in deionized water and in phosphate buffer solution of pH 7.4 at room temperature (25 °C). Figure 6 represents the FT-IR of the degraded samples.

The mechanism of PLA degradation by hydrolysis has been described by many authors. It has been reported that the degradation of PLA-based composites in aqueous medium is caused by autocatalytic hydrolysis of PLA [27]. The ester bonds of PLA are hydrolyzed by water molecules into carboxylic and hydroxyl end groups.
(6)$$R-C\;\left({\mathrm{CH}}_{3}\right){\mathrm{COOR}}^{\prime }+H_{2}O=R-C\;\left({\mathrm{CH}}_{3}\right)\mathrm{COOH}+R^{\prime }\mathrm{OH}$$

This hydrolysis reaction is proportional to time and temperature. At room temperature, PLA hydrolysis is very slow [28].

From Figure 6a, it can be said that for the first 7 days, the degradation of PLA-nHAp nanocomposite occurred very slowly since no peak can be identified at 3340 cm^−1^ and 670 cm^−1^ for the free −OH group. After 15 days of immersion in deionized water, a small change in peak intensity at 670 cm^−1^ and at 3340 cm^−1^ was observed. However, after 30 days, the intensity of peaks at 670 and 3340 cm^−1^ significantly increased due to the presence of −OH group from the hydrolysis of PLA [29].

Figure 6b also showed a similar degradation pattern for the nanocomposite in PBS. However, in PBS, peaks at 664 and 3300 cm^−1^ were noticed after only 7 days instead of 15 days, and significantly increased after 21 days instead of 30 days. So, it can be concluded that, in deionized water, a sudden degradation rate was observed for nanocomposite after 30 days, whereas in PBS (slightly alkaline), a steady degradation rate was observed that started after only 7 days. The steady and comparatively higher degradation rate of nanocomposite in PBS in comparison to deionized water could be due to the alkaline (in PBS) conditions that caused both the surface and bulk degradation of PLA, whereas only bulk degradation occurred for nanocomposite in deionized water [30]. Table 2 represents the change in the pH of the water and the PBS for whole degradation period after every 7 days. For deionized water, the pH value changed; however, it did not follow any linear pattern, whereas for PBS, the pH value was found to decrease with an increase in degradation time.

The change of pH pattern in deionized water can be explained in terms of a high affinity of nHAp for water [31]. In water, the hydrolysis of nHAp predominates causes an increase in pH, and might have initially restricted the hydrolysis of PLA [32]. Later, the hydrolysis of PLA resulted in a decrease in pH. On the contrary, PBS being slightly alkaline in nature (pH = 7.4) caused initially higher pH, and decreased gradually with the hydrolysis of PLA [33].

### 3.8. In-Vitro Cytotoxicity Analysis

Cytocompatibility of the prepared nanocomposite were examined (Appendix A). The findings of the cytotoxicity study are summarized in Table 3. The survival percentage of both the BHK-21 and the Vero cell line were found to be more than 95%. The 5% of the cell that died might be due to the interaction of nanoparticles with cells. As stated in the biological evaluation of medical devices in Part 5, tests for in vitro cytocompatibility (ISO 10993—5:2009) show that any material is cytotoxic if cell viability is less than 70%. From the above statement, it can be said that the prepared nanocomposite is cytocompatible, non-toxic and ideal for biomedical application [34,35].

### 3.9. Drug Loading and In-Vitro Release Profile

#### 3.9.1. Standard Curve of Gentamicin Sulfate

Quantitative analysis of Gentamicin sulfate was performed by generating a standard curve after reacting GS with known concentrations of ninhydrin. Appendix A shows the calibration curve of GS was obtained with 20, 40, 80, 100, 200, 400 and 500 ppm of drug concentrations using the UV–Visible spectrophotometer at a wavelength of 400 nm.

#### 3.9.2. FT-IR Spectra of Drug-Loaded Nanocomposite

GS was loaded in PLA-nHAp nanocomposite via the encapsulation process, and FT-IR spectra were observed before and after drug loading.

Figure 7 represents the comparison of FTIR spectra of the PLA-nHAp nanocomposite and GS-loaded nanocomposite (PLA-nHAp-GS) in the range of 4000–500 cm^−1^. In the FT-IR spectrum of pure GS powder, peaks observed at 2933 cm^−1^ can be assigned for stretching vibrations of N-H amino groups [36]. The typical absorption bands at 1622 and 1528 cm^−1^ can be attributed to the amide I and amide II bond of GS, respectively [21]. Furthermore, a strong absorption band at 1033 cm^−1^ can be attributed to the hydrogen sulfate stretching vibration [37]. The peak observed at 630 cm^−1^ is due to the SO_2_ band [38].

In the drug-loaded nanocomposite (PLA-nHAp-GS), peaks observed at 1100, 1191, 1368, 1455, 1617 and 1752 cm^−1^ resembled the characteristics peaks for the PLA-nHAp nanocomposite and were found to shift a little. Moreover, for the drug-loaded nanocomposite, the characteristic peaks identified at 650 and 2935 cm^−1^ might result from the presence of gentamicin. A newly formed broad absorption peak at 3400 cm^−1^ was observed in PLA-nHAp-GS, which might be due to the stretching vibrations of N–H and OH-O bonds and the intermolecular hydrogen bonding of GS with the polymer matrix [38].

#### 3.9.3. SEM of Drug Loaded Nanocomposite

An SEM image of the drug-loaded nanocomposite (PLA-nHAp-GS) with a corresponding histogram is shown in Figure 8a.

A morphological analysis of the drug-loaded nanocomposite (PLA-nHAp-GS) was carried out to examine if the drug was incorporated properly into the fiber or any structural change occurred after drug loading.

From Figure 8a, smooth and uniform fibers can be observed after drug loading in the nanocomposite by the encapsulation process and is similar to the fiber structure of PLA-nHAp. From the histogram of Figure 8b the average diameter of GS-loaded nanocomposite was found to be 300 nm, whereas for PLA-nHAp, the average diameter was 260 nm. This increase in diameter occurred due to the presence of the drug in the nanocomposite fibers.

#### 3.9.4. In Vitro Drug Release Profile

The drug release behavior of the GS-loaded PLA-nHAp nanocomposite was observed over a period of 8 weeks at 37 °C and pHs of 5.5, 6.0 and 7.4 (Figure 9). As seen from Figure 9, an initial burst release of GS was observed for the first two weeks for all pHs, and this can be attributed to the release of the surface-bounded drug from the nanocomposite [4]. After the initial burst release, a sustained release of the drug was observed throughout the incubation period of 8 weeks with a release of 80%, 70% and 50% at pH 5.5, 6.0 and 7.4, respectively. This slower and sustained release of the drug occurred due to the diffusion of drug from the composite matrix and the swelling and degradation of PLA in the medium [4,39]. This sustained release behavior also indicated the uniform distribution of the drug inside the composite matrix.

Biodegradable polymer-ceramic composite drug release is not only controlled by the degradation of the polymer matrix but also by the diffusion of the drug encapsulated into the polymer matrix [38]. Langer [40] suggested that the drug release from the polymeric matrix includes the migration of drug solutes initially from the polymeric matrix to the polymer’s surface and then to the medium. However, the most influencing factor for drug diffusion is the interaction between the polymer matrix and the incorporated drugs with the release medium [41]. In addition, the pH of the medium has a notable effect on the release rates of gentamicin [4]. In this study, an increased release rate of the drug with reduced pH is observed. Charu et al. [39] also reported the higher release of gentamicin at a lower pH, and suggested that at lower pH (5.5 and 6) there might be a comparatively weaker hydrogen bonding, resulting in the release of greater amount of gentamicin.

#### 3.9.5. Mechanism of Drug Release

The parameters of the distinguished kinetic-release model are elaborated using the equations, where the release data are fitted and the values of correlation coefficients (R^2^) in all conditions analyzed are reported in Table 4. The correlation coefficients are obtained by fitting the drug release data from the drug-loaded electrospun nanocomposites to the kinetic-release models.

It can be said that the Higuchi model appeared to be the best-fitted model for the drug releases from the GS-loaded electrospun nanocomposite (PLA-nHAp-GS) since the correlation coefficient (R^2^) is found to be greater in the Higuchi model than all other kinetic models. This observation suggests that the drug release is controlled by diffusion. The best fit of the Higuchi model confirms the sustained drug release mechanism [42].

From Table 5, the Korsmeyer–Peppas slope exponent (n) was found to be within 0.5 to 0.27 for all the pH media. This also confirms the Fickian release mechanism of GS, meaning that the drug release is followed by diffusion from the PLA-nHAp nanocomposite [42,43].

## 4. Conclusions

In this study, the incorporation of nHAp particles in the PLA matrix was confirmed by FT-IR, XRD and TGA analysis. From the degradation study of nanocomposite in both PBS and water, it can be summarized that degradation in PBS was faster than degradation in deionized water. Cytotoxicity analysis assured nanocomposite’s cell viability was greater than 95%, which confirms its nontoxicity and biocompatibility. Gentamicin sulphate was incorporated in nanocomposite via the encapsulation process. In in vitro analysis, after initial drug release, the gentamicin release was found to be 80%, 70% and 50% at pH 5.5, 6.0 and 7.4, respectively. The drug release from nanocomposite best fit to the Higuchi models confirms the sustained release pattern, and Korsmeyer–Peppas models confirm the Fickian release mechanism. So, it can be concluded that the fabricated drug-loaded electrospun PLA-nHAp nanocomposite has a high potential to be used locally to treat infections by sustained drug release, along with providing nHAp in dental and orthopedic application.

## Figures and Tables

**Figure 1 materials-16-03691-f001:**
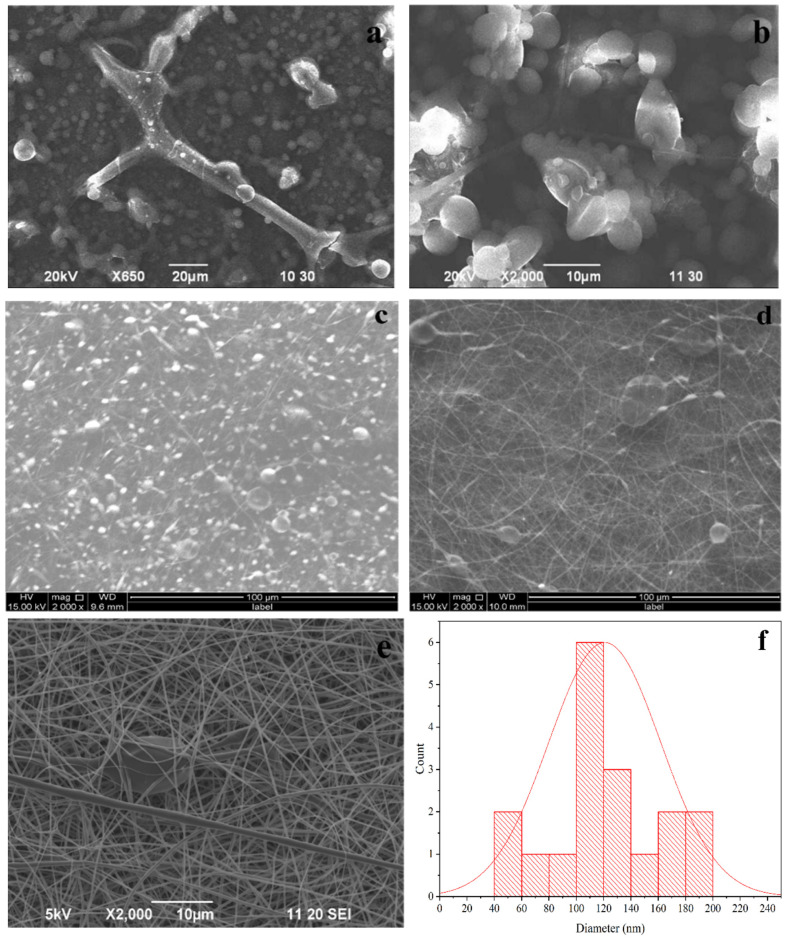
SEM images of (**a**) PF-1, (**b**) PF-2, (**c**) PF-3, (**d**) PF-4, (**e**) PF-5 and (**f**) histogram of PF-5 where the average diameter is found to be 118 nm.

**Figure 2 materials-16-03691-f002:**
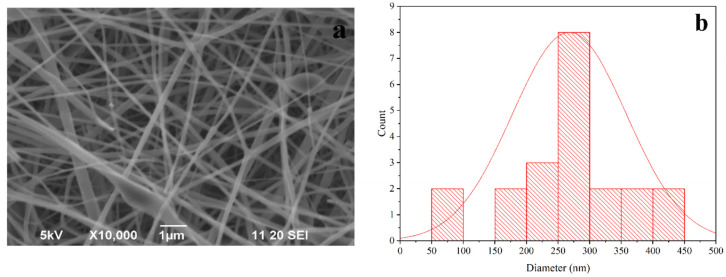
SEM images of PLA-nHAp nanocomposites (**a**) 25:3 and (**b**) histogram of PLA-nHAp (25:3).

**Figure 3 materials-16-03691-f003:**
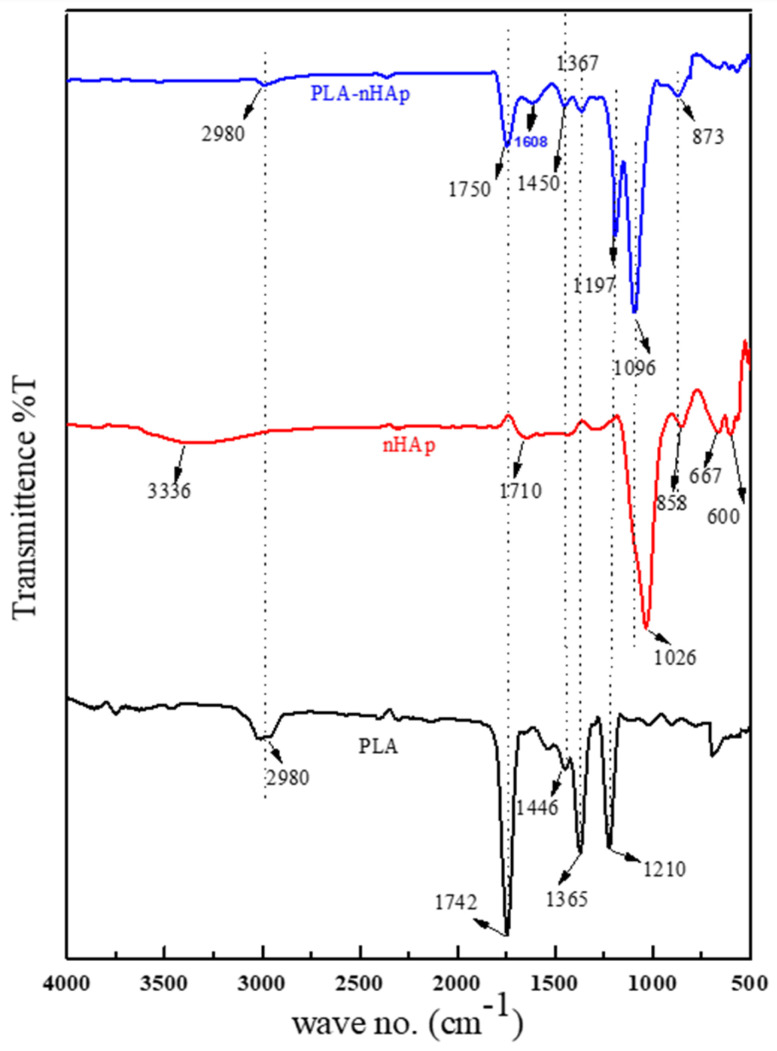
FT-IR spectrum of PLA, nHAp and PLA-nHAp.

**Figure 4 materials-16-03691-f004:**
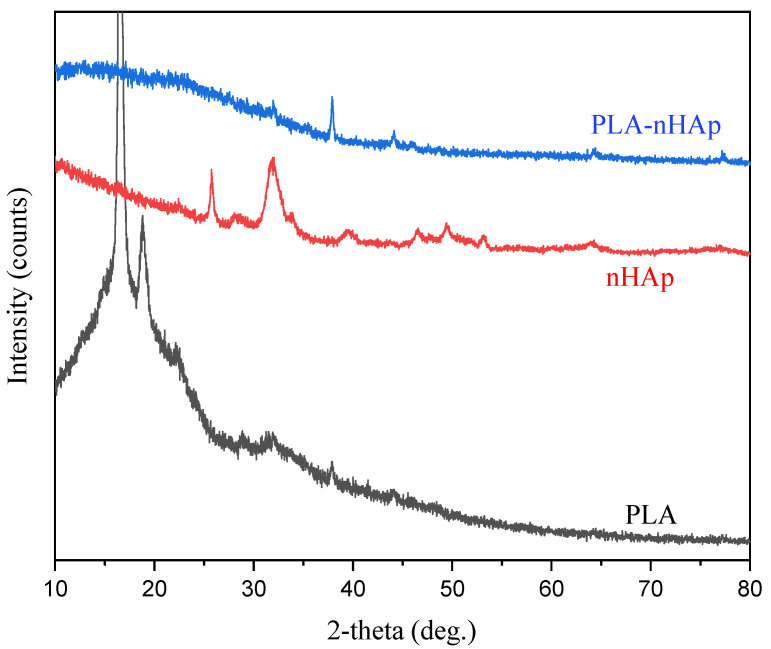
X-ray diffractogram of PLA, nHAp and PLA-nHAp.

**Figure 5 materials-16-03691-f005:**
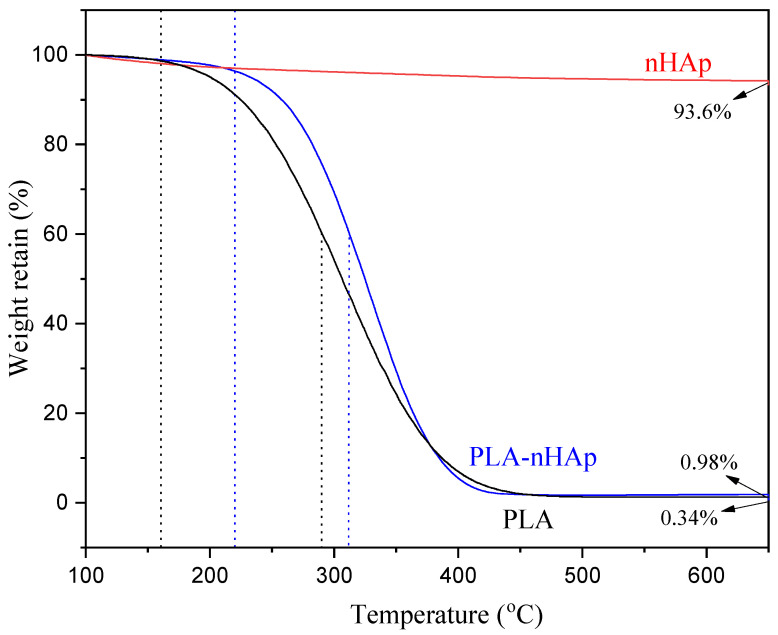
TGA thermogram of PLA, nHAp and PLA-nHAp.

**Figure 6 materials-16-03691-f006:**
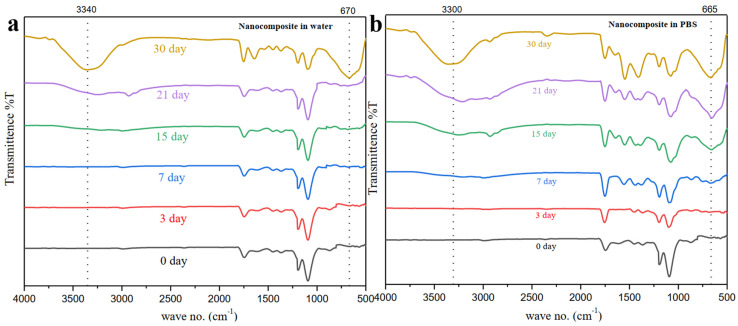
FT-IR of degradation study of PLA-nHAp nanocomposite in (**a**) water and (**b**) PBS.

**Figure 7 materials-16-03691-f007:**
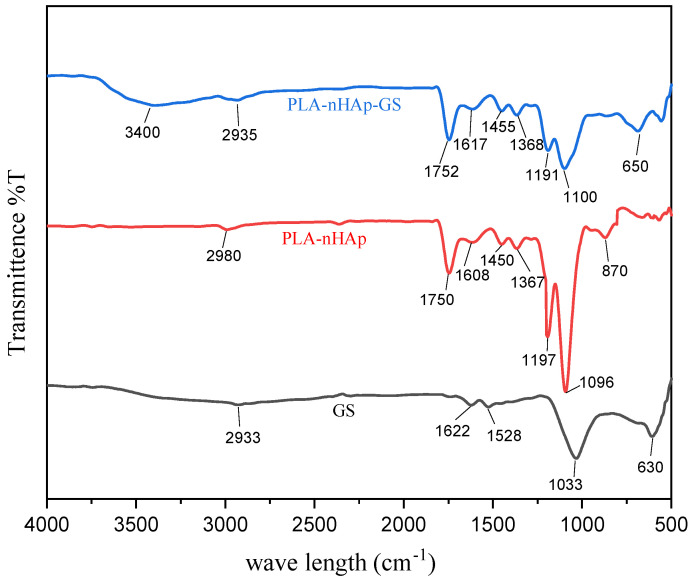
FTIR of PLA-nHAp, GS and PLA-nHAp-GS.

**Figure 8 materials-16-03691-f008:**
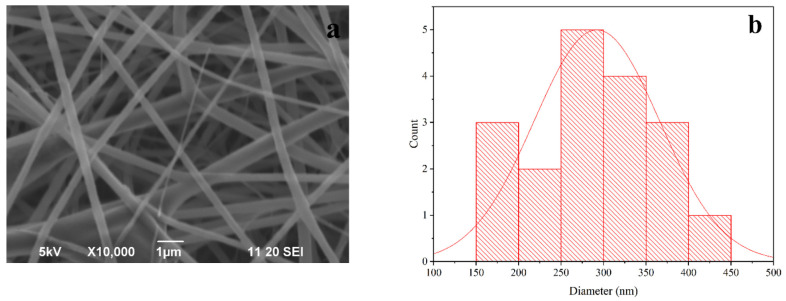
(**a**) SEM image and (**b**) histogram of Gentamicin (GS)-loaded PLA-nHAp nanocomposite.

**Figure 9 materials-16-03691-f009:**
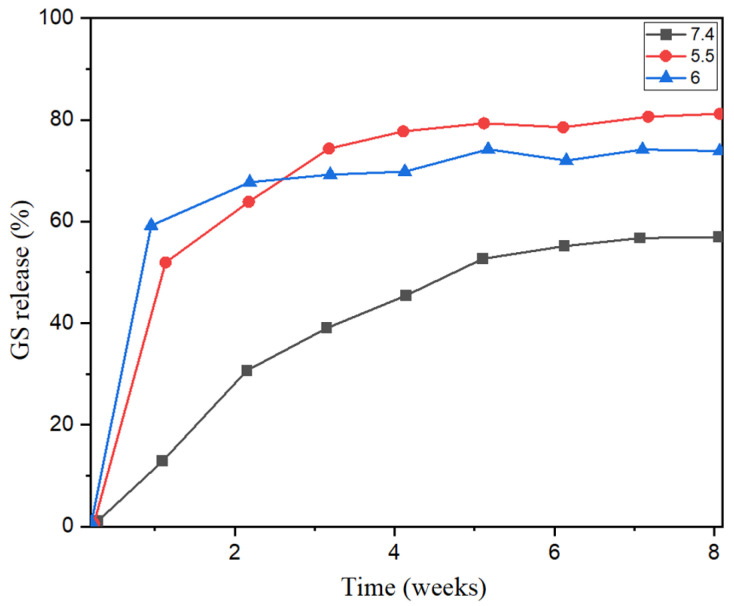
Drug release profile of drug-loaded nanocomposite in phosphate buffer solution of pH 5.5, 6 and 7.4.

**Table 1 materials-16-03691-t001:** Composition of PLA fiber.

Sample	Composition (Concentration, %*w*/*v*)
PF-1	5%
PF-2	10%
PF-3	15%
PF-4	20%
PF-5	25%

**Table 2 materials-16-03691-t002:** Change in pH value during degradation for PLA-nHAp both in deionized water and PBS.

Duration	pH Value	
	Water	PBS
0 days	6.13	7.40
7 days	6.60	7.28
14 days	7.40	7.13
21 days	7.04	7.07
30 days	6.70	6.96

**Table 3 materials-16-03691-t003:** Cytotoxicity study of the prepared PLA-nHAp nanocomposite.

Sample ID	BHK-21 Cell’s Survival Rate	Vero Cell’s Survival Rate	Remarks
Control	100%	100%	---
PLA-nHAp nanocomposite	>95%	>95%	No cytotoxicity

**Table 4 materials-16-03691-t004:** The correlation coefficients obtained by fitting the drug release data from the drug-loaded electrospun nanocomposites to the kinetic-release models.

Correlation Coefficient (R^2^)			
Kinetic Models	pH 7.4	pH 6	pH 5.5
Zero-order	0.9369	0.6763	0.7680
First-order	0.9606	0.8586	0.8613
Higuchi	0.9848	0.9198	0.9382
Korsmeyer-Peppas	0.9807	0.8260	0.8939

**Table 5 materials-16-03691-t005:** The Korsmeyer–Peppas slope exponent (n) found from the release data.

Media	Value of n	Drug Transport Mechanism
pH 7.4	0.5	Fickian diffusion
pH 6	0.27	Quasi Fickian
pH 5.5	0.32	Quasi Fickian

## Data Availability

Not applicable.

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
