# Peer review of "Fabrication of Electrospun PLA-nHAp Nanocomposite for Sustained Drug Release in Dental and Orthopedic Applications"

_materials, 2023, doi:10.3390/ma16103691_

Round 1
Reviewer 1 Report
Comments:
The manuscript reports on the “fabrication of electrospun PLA-nHAp nanocomposites for sustained drug release in dental and orthopedic applications”. The topic is timely and important, but the manuscript has several shortcomings that preclude its acceptance for publication.
1. The introduction section lacks clarity and concision, and the citations are not up to date. The authors should provide a clear and concise introduction to the study and highlight the most relevant and recent literature on PLA-nHAp nanocomposites for drug delivery. The references cited should be recent and pertinent to the research presented in the manuscript.
2. The Turnitin report attached to the manuscript indicates a need for further work to reduce plagiarism and provide more justification for the results and novelty of the work presented. The authors should carefully review the manuscript and ensure that all sources are appropriately cited and any passages that may be deemed problematic are rephrased.
3. The references and discussion sections require improvement. The authors should provide more thorough justification and clarification of the trends identified in the literature and explain how the addition of nHAp affects the properties of PLA. The discussion should be more focused, and the results should be more clearly linked to the research question.
4. The diagrams included in the manuscript require proper labeling and appropriate use of software to represent the results. The authors should ensure that all diagrams and figures are clearly labeled and cited in the text. The diagrams should be of high quality, and the software used to generate them should be appropriate for academic publications.
5. It would be helpful if the authors could discuss the implications of their findings for future research in PLA-nHAp nanocomposites for sustained drug delivery in dental and orthopedic applications. The authors should provide insights into the potential applications of their research and how it may be extended or built upon in future studies.
6. In conclusion, while the topic of the manuscript is of significant interest, the current state of the manuscript falls short of the standards required for publication. The authors are encouraged to revise their manuscript to address the issues identified and to resubmit it for consideration in the future.

Reviewer 2 Report
Comments
Quantitative information should be provided in the abstract section.
The author should improve with relevant citations in the introduction.
In section 2.2. Nano-hydroxyapatite (nHAp) synthesis. Citation need.
This manuscript found in term “Error! Reference source not found”. The author should correct it in the throughout manuscript.
Equation should be formatted.
The conclusion section provided with outstanding point of this work.
Typographical errors must be corrected throughout the manuscript (i.e, superfluous spaces, inconsistent use of units, superscript, etc.).
Supporting Information file is missing in this submission.
Reviewer 3 Report
Dear Authors,
The article is generally cool. There are some minor deficiencies related to editing and a bit with the intro. Correct this then I recommend the article for acceptance.
In terms of editing
- names of reagent suppliers;
- CAS numbers;
- tube length for XRD;
- type of detector for SEM;
- add process flow chart;
- you have a lot of omissions in the text - see how many "error" messages there are;
- better describe electro-spinning.
Intro/methodology
- larger SEM photos + photos with detector description (plus extra current and voltage);
- in the intro, write something extra about alternative methods of preparing composites; read it and quote:
(1)
Influence of irradiation parameters on the curing and interfacial tensile strength of HAP printed part fabricated by SLA-3D printing
(2)
Alkali-Treated Alumina and Zirconia Powders Decorated with Hydroxyapatite for Prospective Biomedical Applications
The rest is fine, nothing to complain about. Just correct what I have mentioned.
Best Wishes
Reviewer
Round 2
Reviewer 1 Report
Recommended for acceptance.